# Glycation of Host Proteins Increases Pathogenic Potential of *Porphyromonas gingivalis*

**DOI:** 10.3390/ijms222112084

**Published:** 2021-11-08

**Authors:** Michał Śmiga, John W. Smalley, Paulina Ślęzak, Jason L. Brown, Klaudia Siemińska, Rosalind E. Jenkins, Edwin A. Yates, Teresa Olczak

**Affiliations:** 1Laboratory of Medical Biology, Faculty of Biotechnology, University of Wrocław, 14A F. Joliot-Curie St., 50-383 Wrocław, Poland; michal.smiga@uwr.edu.pl (M.Ś.); paulina.stepien2@uwr.edu.pl (P.Ś.); klaudia.sieminska@uwr.edu.pl (K.S.); 2Institute of Life Course and Medical Sciences, School of Dentistry, The University of Liverpool, Pembroke Place, Liverpool L3 5PS, UK; josmall@liv.ac.uk (J.W.S.); Jason.Brown@glasgow.ac.uk (J.L.B.); 3CDSS Bioanalytical Facility, Department of Pharmacology and Therapeutics, Institute of Systems, Molecular and Integrative Biology, Faculty of Health and Life Science, The University of Liverpool, Liverpool L69 3GE, UK; R.Jenkins@liv.ac.uk; 4Institute of Systems, Molecular and Integrative Biology, Faculty of Health and Life Science, The University of Liverpool, Liverpool L69 7ZB, UK; e.a.yates@liv.ac.uk

**Keywords:** periodontitis, diabetes, glycation, *Porphyromonas gingivalis*, heme, HmuY, hemoglobin, albumin, collagen, pathogenesis

## Abstract

The non-enzymatic addition of glucose (glycation) to circulatory and tissue proteins is a ubiquitous pathophysiological consequence of hyperglycemia in diabetes. Given the high incidence of periodontitis and diabetes and the emerging link between these conditions, it is of crucial importance to define the basic virulence mechanisms employed by periodontopathogens such as *Porphyromonas gingivalis* in mediating the disease process. The aim of this study was to determine whether glycated proteins are more easily utilized by *P. gingivalis* to stimulate growth and promote the pathogenic potential of this bacterium. We analyzed the properties of three commonly encountered proteins in the periodontal environment that are known to become glycated and that may serve as either protein substrates or easily accessible heme sources. In vitro glycated proteins were characterized using colorimetric assays, mass spectrometry, far- and near-UV circular dichroism and UV–visible spectroscopic analyses and SDS-PAGE. The interaction of glycated hemoglobin, serum albumin and type one collagen with *P. gingivalis* cells or HmuY protein was examined using spectroscopic methods, SDS-PAGE and co-culturing *P. gingivalis* with human keratinocytes. We found that glycation increases the ability of *P. gingivalis* to acquire heme from hemoglobin, mostly due to heme sequestration by the HmuY hemophore-like protein. We also found an increase in biofilm formation on glycated collagen-coated abiotic surfaces. We conclude that glycation might promote the virulence of *P. gingivalis* by making heme more available from hemoglobin and facilitating bacterial biofilm formation, thus increasing *P. gingivalis* pathogenic potential in vivo.

## 1. Introduction

Periodontitis belongs to a group of multifactorial, non-communicable infectious diseases characterized by inflammation and the destruction of tooth-supporting tissues. An altered or disturbed mutualism between oral microbiome members results in dysbiosis with local injury and subsequently, in systemic diseases [1,2,3,4,5,6,7]. There is growing evidence that a two-way relationship exists between periodontitis and diabetes: diabetes increases the risk of periodontitis and periodontitis is associated with compromised glycemic control [8,9,10,11,12]. Diabetic patients, mostly those displaying poor glycemic control, experience severe forms of gingivitis and are at a high risk of periodontitis onset and progression [13,14,15]. The pathogenic process that links these two diseases results from upregulated inflammation arising from each condition exerting mutually adverse effects [16]. Diabetes increases the risk of periodontitis by contributing to increased inflammation in the periodontal tissues, leading to the activation of local immune and pro-inflammatory responses, local tissue damage, increased breakdown of the periodontal connective tissues and resorption of alveolar bone and the exacerbation of periodontitis. Periodontopathogens and their products, together with inflammatory cytokines and other locally produced mediators in the inflamed periodontal tissues, enter circulation and contribute to up-regulated systemic inflammation. This leads to impaired insulin signaling, insulin resistance and the exacerbation of diabetes [17,18].

Diabetic patients suffer from a variety of systemic health problems, resulting primarily from glycated circulatory and tissue proteins. These arise from the non-enzymatic reaction between amino groups and reducing sugars, resulting mainly in the addition of glucose moieties to the proteins. This chemical modification involves the Maillard reaction, forming first the Schiff base, which undergoes rearrangement to an N-linked (glycosylamine) product. Further reactions follow, via Amadori rearrangement, to ketosamines and resulting finally in increased levels of Advanced Glycation End (AGE) products [19,20]. The main target of these glycation reactions is the amino group of lysine residues, which reduces the availability of these side chains in proteins [21]. It has been demonstrated that glycation may change both structure and function, with profound effects on both the biochemical and physiological properties of such proteins. For instance, the glycation of albumin decreases its ability to bind and carry ligands [22,23,24]. In experimental animals, the half-life of circulating glycated albumin is decreased and glycated albumin is degraded about 33% faster than the un-glycated protein [22]. Glycated hemoglobin shows increased oxygen binding and decreased oxygen unloading [25], a factor that may contribute to the anaerobic nature of the periodontal tissues and periodontal pocket. Other serious effects of hemoglobin glycation involve the promotion of thermal instability, oxidation to the methemoglobin form and, importantly, the induction of a loss of iron and heme transfer from glycated hemoglobin subunits to serum albumin [26,27].

Clinical studies have demonstrated that there is increased colonization by “red complex” microorganisms (*Porphyromonas gingivalis*, *Tannerella forsythia*, *Treponema denticola*) in the subgingival plaque of individuals with type 2 diabetes [14]. Moreover, the increased numbers of black pigmented anaerobes at such sub-gingival sites also correlates with disease severity and the raised levels of glycated proteins not only in serum, but also in gingival crevicular fluid [14,15,28,29,30]. It has been reported that the gingival crevicular fluid from the periodontal lesions of diabetic patients contains both glycated albumin and glycated hemoglobin [31], and that elevated levels of glycated hemoglobin are associated with an increased inflammatory status and correlates with the severity of periodontitis in diabetic patients [28]. One may speculate that the elevation of these glycated hemoproteins may represent a more efficient nutrient supply for periodontopathogens, which could result in the stimulation of the growth and/or pathogenic potential of subgingival microorganisms. One other protein, which is also glycated in vivo, is collagen, the most abundant, long-life structural protein in the human body [29,32,33]. This protein has a hierarchical arrangement originating from the polymerization of collagen molecules into fibrils and, owing to its abundant lysine residues, is also subjected to intermolecular crosslinking from glycation [29,32,33,34,35,36].

It is also noteworthy that *P. gingivalis*, a keystone etiologic agent of chronic periodontitis, is more frequently detected in periodontal pockets of diabetic patients with raised glycated hemoglobin (HbA_1c_) levels [30], and that increased gingival tissue bleeding is seen in diabetic individuals [37,38]. This raises the question as to whether heme and iron availability shape the microbiological profile in the plaque of diabetic patients. In this context, it is important to note that in order to survive in the host environment, and initiate and progress periodontitis, black-pigmented anaerobes, such as *P. gingivalis*, must acquire heme as a sole source of iron and protoporphyrin IX [39]. Moreover, heme is an important key virulence factor for *P. gingivalis*.

Our previous work has resulted in an extensive biochemical and functional characterization of one of the major *P. gingivalis* heme uptake systems (Hmu), has progressed our understanding of the mechanisms of heme acquisition and revealed the leading role played by the TonB-dependent outer membrane receptor HmuR and hemophore-like HmuY proteins [40,41,42,43,44,45,46,47,48]. Specifically, we demonstrated that HmuR is involved in heme transport through the *P. gingivalis* outer membrane, whereas HmuY is a unique heme-binding protein, transferring heme to the HmuR. These proteins serve as the first documented example of a novel, two-component system, comprising an outer-membrane transporter and a heme-binding hemophore-like protein. We have also shown that *P. gingivalis* displays a novel heme acquisition paradigm from hemoglobin, whereby oxyhemoglobin must be first oxidized to methemoglobin, which facilitates heme release. In the case of *P. gingivalis*, this process involves the arginine-specific gingipain protease A (HRgpA) [48]. The bacterium is then able to fully proteolyze the more susceptible methemoglobin to release free heme, mostly through the proteolytic activity of *P. gingivalis* lysine-specific gingipain K (Kgp) and via heme sequestration by HmuY. *P. gingivalis* HmuY can capture heme directly from methemoglobin [48] and thus functions similarly to classical, bacterial hemophores, which are secreted proteins engaged in heme transfer from the host hemoproteins to the outer-membrane receptors [49]. HmuY is also able to compete with albumin, which is the normal front-line circulatory heme scavenger in vivo [48], as well as acquiring heme from serum hemopexin [47].

The HmuY protein is associated with the outer membrane of the bacterial cell and outer-membrane vesicles (OMVs) through the lipid anchor [50,51,52], and can be shed in an intact, soluble form through the limited proteolytic activity of *P. gingivalis* Kgp [45,50]. This is especially important since not only whole *P. gingivalis* cells but also OMVs can be internalized into endothelial and epithelial cells via an actin- and lipid raft-mediated endocythic pathway and, therefore, have been shown to play a role in establishing colonization, carrying and transmitting virulence factors and modulating the host response [53,54,55]. From this perspective, their cargo components, such as the well documented gingipain proteases and HmuY [50,51,52,56,57,58], appear to be important both for *P. gingivalis* colonization and virulence. Moreover, it has been demonstrated that *P. gingivalis* residing in the oral cavity can translocate to liver cells, causing decreased insulin sensitivity [59]. Importantly, mouse model experiments have revealed that hepatic cells did not harbor *P. gingivalis* cells, but the contents of OMVs led to the attenuation of insulin sensitivity and the inhibition of hepatic glycogen synthesis [18].

Given the high incidence of periodontitis and diabetes and the emerging link between these conditions, defining the basic virulence mechanisms employed by *P. gingivalis* is of crucial importance in understanding these microbially mediated disease processes. *P. gingivalis*, selectively enriched by growth in gingival crevicular fluid, is able to efficiently degrade the serum iron- and heme-scavenging proteins, which otherwise defend the gingival crevice against microbes by sequestering any free iron and heme [60,61]. However, little is known regarding the influence that glycated host proteins within the gingival crevice, periodontal pocket and the tissues of the periodontium have upon the growth and pathogenic potential of *P. gingivalis*. Therefore, the general purpose of this study was to determine whether glycated proteins are more easily utilized by *P. gingivalis*, whether they have a growth stimulatory effect, and whether they could promote the pathogenic potential of this bacterium.

## 2. Results and Discussion

### 2.1. Characterization of Glycated Proteins

Due to the difficulties obtaining sufficient standardized glycated hemoglobin from diabetic patients, we employed horse hemoglobin, which was glycated in vitro by incubation with glucose, using a modified protocol previously described by Ch’ng and Marinah [62]. The glycated moieties were further purified from the non-modified proteins using affinity chromatography, and the samples were analyzed using colorimetric analysis, sodium dodecyl sulfate-polyacrylamide gel electrophoresis (SDS-PAGE) and liquid chromatography/mass spectrometry (LC/MS).

The efficiency of glycation was confirmed using an assay for the presence of 5-hydroxymethylfurfural (5-HMF) using the thiobarbituric acid method [63]. From three separate determinations, we found that the average molar ratio of 5-HMF to hemoglobin subunit was 1.2:1, indicating that each protein subunit was minimally modified by one hexose unit (Appendix A). It is noteworthy that 5-HMF was also detected in the un-glycated hemoglobin samples, likely arising from the natural in vivo glycation of hemoglobin in normal individuals [64]. The level of glycation in the un-glycated hemoglobin preparations was relatively low, with the average molar ratio of 5-HMF to hemoglobin being 0.2, indicating one modification per five hemoglobin subunits (Appendix A).

SDS-PAGE analysis (Figure 1a) demonstrated the slower migration of the glycated compared to un-glycated hemoglobin, confirming the increased molecular weight due to the hexose modifications of the protein. Both hemoglobin samples contained almost equal amounts of methemoglobin (95% for glycated and 97% for un-glycated samples); the remaining component in both preparations being hemichrome.

LC/MS analysis of the glycated hemoglobin used in our study showed a very similar pattern of glycation to human hemoglobin both in vivo and in vitro, which is in accordance with the published data (Appendix A and Appendix A) [65,66,67,68]. Interestingly, weak signals were also observed on non-lysine sites in the α chain (e.g., on serine at positions 3 and 84 and on threonine at positions 39 and 137). A study of the reaction of formaldehyde [69] found evidence for a reaction with serine and threonine, and the finding here of modifications on both amino acids is consistent with these residues having reacted similarly with the reducing end (an aldehyde group) of the sugar. Importantly, glycated residues Lys61(α), Lys90(α), Lys66(β) and Lys95(β) were positioned in the heme pocket, with Lys61(α) and Lys66(β) located on the E helix of the α- and β-chains, which contain the histidine residues (His58 and His63, respectively) involved in oxygen binding to the heme iron, whilst Lys90(α) and Lys95(β) are positioned on the same F8 helix-loop as the proximal histidines (His87(α) and His92(β), respectively), which are essential in coordinating the heme iron to the hemoglobin peptide chains.

Circular dichroism (CD) spectra demonstrated subtle changes in the secondary structure of hemoglobin after glycation as compared to the un-glycated protein (Figure 2a; shown by arrows). The UV–visible spectra did not reveal significant changes between un-glycated and glycated hemoglobin (Figure 2b,c). However, the glycated hemoglobin fraction exhibited a greater tendency to release iron compared to the un-glycated protein, as determined using the ferrozine assay method (Appendix A), which is in accordance with published data demonstrating results from both in vivo and in vitro glycation [26,70].

According to the manufacturer’s product specification (A8301, Sigma-Aldrich, St. Louis, MO, USA) glycated albumin has between 1 and 5 mol of hexose (determined as fructosamine) per mol of protein.

An assessment of the degree of glycation revealed that the glycated collagen comprised 30.23 ± 0.48 (n = 3) nmols of 5-HMF per mg of protein, whilst the non-glycated collagen control sample contained 1.45 ± 0.09 (n = 3) nmols of 5-HMF per mg of protein. These values are within the ranges of the 5-HMF levels found in the insoluble collagens extracted from both human and animal species [71]. Based on this assay, we calculated that each type I tropocollagen molecule glycated under the above conditions was modified (to the nearest whole number) by nine glucose additions. This value is similar to the predicted number of lysine residues (13) in type I tropocollagen, which are susceptible to glycation [72].

Similar to hemoglobin, both the glycated albumin (Figure 1b) and glycated collagen fractions (Figure 1c) migrated slightly slower during SDS-PAGE compared to their un-glycated counterparts, providing evidence of the addition of glucose to the proteins.

### 2.2. Glycated Hemoglobin as a Sole Source of Heme for P. gingivalis

Despite changes in the structure and properties of glycated proteins, and the fact they are ubiquitously formed and present in periodontal vascular and connective tissues, gingival crevicular fluid and periodontal pockets during periods of hyperglycemia, nothing is known regarding their influence on the growth and virulence of the sub-gingival periodontopathogens. Therefore, we determined any possible growth-promoting ability of glycated hemoglobin, whether it might be degraded more readily by *P. gingivalis* for the purposes of heme release, or if heme might be more easily extracted by the HmuY hemophore-like protein for transfer to HmuR at the bacterial surface.

Rgp and Kgp gingipains act synergistically to degrade hemoglobin, which results in the release of heme for cellular uptake and the production of pigment through its cell-surface deposition in the form of a μ-oxo heme dimer [73,74]. Although Rgp gingipains have lower proteolytic activity toward hemoglobin compared to Kgp, HRgpA is efficient in converting oxyhemoglobin into methemoglobin, the latter being more easily attacked by Kgp [75] and from which heme can be sequestered by HmuY [48]. Therefore, we first examined whether glycated hemoglobin, compared to the un-glycated protein, was a more susceptible substrate for gingipains. However, as shown in Figure 3, although no significant differences were observed in protease susceptibility in *P. gingivalis* cultures between the two hemoglobin preparations, it seems that un-glycated hemoglobin is a slightly better substrate for *P. gingivalis* proteases. To explore this further, we employed gingipain mutant strains. Our results showed a similar trend as compared with the wild-type strain, apart from lower proteolytic activity resulting from the lack of genes encoding respective gingipains (Figure 3). These results also confirmed that among all the proteases produced by *P. gingivalis*, the gingipain fraction was mainly responsible for hemoglobin degradation.

The three-dimensional rendering of the horse methemoglobin molecule (PDB ID: 2MHB) [76] (Appendix A) revealed that the glycated lysine residues were mainly surface located (Lys16(α), Lys8(β), Lys82(β) and Lys144(β)), and it is likely that this would slightly reduce the efficiency of the gingipain degradation of the glycated protein by blocking the Kgp target residues. Additionally, when we incubated hemoglobin samples with *P. gingivalis* cells, it was revealed that heme was released from the un-glycated hemoglobin more easily compared to the glycated protein (Figure 4). Nevertheless, we found that the addition of glycated hemoglobin to basal medium (BM) as a sole source of heme resulted in a slightly, but not statistically significant increased growth of *P. gingivalis* compared to that in the presence of the un-glycated protein (Figure 5). This shows that despite the fact that glycated hemoglobin is degraded slightly less efficiently by *P. gingivalis* proteases, it could be used as a heme source as efficiently as un-glycated hemoglobin.

One notable observation was the difference in the ability of the HmuY to sequester heme from the glycated and un-glycated hemoglobin (Figure 6). Specifically, the HmuY-Fe(III)-heme complex formed more rapidly from the glycated than from un-glycated hemoglobin (relative heme binding constant for HmuY of 16.6 ± 1.0 × 10^−3^ s^−1^ from a glycated sample compared to 11.3 ± 0.7 × 10^−3^ s^−1^ for the un-glycated sample). Thus, although both hemoglobin preparations contained almost equal concentrations of methemoglobin, there was an approximately 1.5-fold increase in the rate of heme acquisition. These data suggest that glycated hemoglobin is a more ready substrate as a source of heme than un-modified hemoglobin. The far-UV CD analysis showed that both glycated and un-glycated hemoglobin shared a high similarity in secondary structure, indicating that the α-helix strands (180–200 nm peak) and β-sheets (200–240 nm trough) were largely unaffected by the glycation process (Figure 2b,c). We can conclude that the in vitro glycation process did not result in the unfolding of the protein, a finding contrary to the work of Sen et al. [26], but this may be explained by the fact that structural changes can occur during longer periods of glycation. In the near-UV region, however, there were subtle differences between the two spectra: the peak intensity between 260 and 280 nm was increased for the glycated hemoglobin (Figure 2a). In addition, a small change was seen in the shoulder at 270 nm in the un-glycated hemoglobin sample, which was less prominent than in the glycated hemoglobin. Taken together, these spectra indicated that whilst the secondary structure was unaffected, in vitro glycation appeared to alter the spatial disposition of the aromatic amino acids, e.g., the orientation of the phenylalanine (260–270 nm) and/or tryptophan (275–285 nm region) residues. Moreover, the glycation of lysine residues located in the hemoglobin heme-binding pocket may result in structural changes in this region, resulting in an increased accessibility of heme for HmuY. Indeed, it is known that the heme-globin linkage is weakened in both glycated hemoglobin and myoglobin, which increases the rate of heme transfer to serum albumin [26,77]. Whilst as yet we cannot provide an exact mechanism for the increase in heme extraction from glycated hemoglobin by HmuY, it is notable that Lys61(α) and Lys66(β) were found to be glycated, which is in keeping with other work [78,79]. These residues lie within the heme pocket on the E helix of the α- and β-chains, carrying the residues His58 and His63, which control oxygen binding to the heme iron. As these lysine residues cooperate with the histidine in oxygen binding, it is likely that glycation at these sites may not only reduce oxygen affinity [80,81], but destabilize the heme to protein interactions. In addition, it is known that glycated oxyhemoglobin and oxymyoglobin both have an increased propensity to auto-oxidize to their met-forms [26,77]. This may influence heme extraction by HmuY, since methemoglobin is its preferred substrate [39], although it should be noted that both glycated and un-glycated hemoglobin had similar levels of methemoglobin (95 and 97%, respectively).

Based on our results, we assumed that although there were differences in hemoglobin degradation and heme release and/or heme extraction between glycated and un-glycated hemoglobin, these might not be sufficient to impart a major advantage for *P. gingivalis* growth in liquid culture media under laboratory conditions. Important features of *P. gingivalis*-mediated diseases include the ability of the bacterium to infect and disseminate through host cells and tissues and subvert host immunological surveillance and defense mechanisms [1,53,82]. Therefore, we examined how *P. gingivalis* infected host cells cultured in the presence of hemoglobin as a main sole source of heme. We found that the glycation of hemoglobin slightly increased the general infection ability towards cultured keratinocytes (Figure 7). This effect was mainly caused by the slightly higher tendency of bacteria to attach to, rather than to invade, the keratinocytes. Based on these results, one may assume that glycation might improve the ability of *P. gingivalis* to survive and subsequently migrate into deeper layers of the epithelium using the intercellular pathway rather than the intracellular one. It is worth mentioning, however, that differences between cultures carried out in the presence of un-glycated and glycated hemoglobin were not statistically significant. One of the explanations of this effect could be the employment in our experiments of a simple model composed of keratinocytes cultures, which did not allow us to observe the invasion process performed under in vivo conditions. A recent study, based on the vascularized three-dimensional gingival model with an epithelial barrier expressing cell–cell junctions using collagen microfibers, demonstrated such a possibility [83].

### 2.3. Glycated Albumin as a Sole Source of Heme for P. gingivalis

It has been demonstrated that the environment of sub-gingival dental plaque, which is enriched in serum albumin as a result of inflammatory processes, selects for the presence and encourages the growth of black-pigmented anaerobes, including *P. gingivalis* [60]. This may have the effect of increasing periodontal tissue destruction at such sites, since this is accompanied by increased collective proteolytic activity of the organisms present. In view of this, we examined whether glycated albumin, compared to the un-glycated protein, might serve as a more susceptible substrate for gingipain activity. However, similar to hemoglobin, as shown in Figure 8, no significant differences were observed in the susceptibility of the two albumin species to proteolysis in *P. gingivalis* cultures.

Unlike hemoglobin, we found only a slight increase in heme sequestration by HmuY from glycated versus un-glycated albumin (Appendix A). Shaklai et al. [22] showed that the glycation of human serum albumin does not alter heme binding. We also showed that both the un-glycated and glycated albumin exhibited similar UV–visible spectra, demonstrating a similar heme-binding efficiency (Appendix A). This can be explained by the fact that the main glycation site (Lys525) [84] and the heme binding site (His146 and Tyr161) are not in close proximity within the albumin molecule (Appendix A). This is in contrast to glycated hemoglobin, where a relaxation of heme binding occurs [26].

### 2.4. Glycated Collagen as a Substratum for P. gingivalis

Collagen is the most abundant, long-lived structural protein and the main component of the basal structures of connective tissues [85]. In the oral cavity, type I collagen is an important matrix constituent of the hard and soft tissues, including dental pulp, dentine, periodontal ligament and alveolar bone. Type I collagen comprises 70% of the total collagen in periodontal ligament, where it forms a highly cross-linked matrix. To partially mimic this substrate, we used type I collagen derived from a bovine Achilles tendon, which was glycated by incubation with glucose.

It has been suggested that the glycation of collagen may promote bacterial adhesion, and even penetration into the collagen gel subsurface, resulting in subsequent biofilm formation. A recent report showed that glucose-induced crosslinking of collagen increased biofilm formation by *Streptococcous mutans*, but not by *Streptococcus sanguinis* [59]. Members of the oral microbiome can also utilize type I collagen as a substratum for both attachment and biofilm formation [86]. For this purpose, *P. gingivalis* uses several proteins, including RgpA and adhesins [87,88,89]. As shown in Figure 9, the glycation of collagen increased the tendency for biofilm formation by *P. gingivalis* on surfaces coated with the modified protein compared to those coated with un-glycated collagen, as well as to surfaces coated with bovine serum albumin, and un-coated abiotic surfaces.

We did not find any differences between un-glycated and glycated collagens in terms of protein degradation by *P. gingivalis* proteases (data not shown). Due to the large number of lysine residues potentially available for glycation in collagen, one could predict it to be a less amenable source of amino acids for *P. gingivalis* due to blocking the access of Kgp to cleavage sites even when the native triple helix is unwound. However, although we did not find a difference in digestion between glycated and un-glycated collagen in our in vitro assay, one may not rule out the possibility that glycated collagen, compared to its un-glycated counterpart, could be subjected to more efficient destruction in situ within the periodontal tissue matrix. One of the possible explanations is the more complex structure of gingival tissue composed of an epithelial cell layer and a dense fibrillar connective tissue, the latter comprising mostly type I collagen.

## 3. Materials and Methods

### 3.1. Bacterial Stains and Growth Conditions

*P. gingivalis* W83 wild-type and Δ*kgp*, Δ*rgpA*Δ*rgpB*, Δ*kgp*Δ*rgpA*Δ*rgpB* gingipain mutant strains, lacking respective functional genes, constructed in the W83 genetic background [90,91,92], were grown anaerobically (Whitley A35 anaerobic workstation; Bingley, UK) at 37 °C for 5 days on blood agar plates composed of Schaedler broth (containing hemin and L-cysteine), and supplemented with 5% sheep blood and menadione (Bimaxima, Lublin, Poland). Mutant cells were grown with an addition of appropriate antibiotics. These cultures were inoculated into liquid basal medium (BM) comprising 3% trypticase soy broth (Becton Dickinson, Sparks, MD, USA), 0.5% yeast extract (Biomaxima), 0.5 mg/L menadione (Fluka, Munich, Germany) and 0.05% L-cysteine (Sigma-Aldrich, St. Louis, MO, USA). To grow bacteria under high iron/heme conditions (BM + Hm), BM was supplemented with 7.7 μM hemin (Fluka). To grow bacteria under low iron/heme conditions (BM + DIP), hemin was not added to BM and iron was chelated by an addition of 160 μM 2,2-dipyridyl (DIP; Sigma-Aldrich). To assess the ability of *P. gingivalis* to grow in BM, where the source of heme was 2 μM glycated or un-glycated hemoglobin, bacteria were grown in BM + Hm for 24 h, followed by 2 passages in BM + DIP, starting at optical density at 600 nm (OD_600_) of at least 0.25. Then, bacteria were centrifuged (4000× *g*, 20 min, 4 °C), washed once with 20 mM sodium phosphate buffer, pH 7.4, supplemented with 140 mM NaCl (PBS), and again centrifuged. These samples were used to inoculate fresh BM + Hb culture media at an initial OD_600_ of 0.2. The growth was determined by measuring the OD_600_ over time.

*Escherichia coli* Rosetta (DE3)RIL strain (Agilent Technologies, Santa Clara, CA, USA) was cultured under standard aerobic conditions.

### 3.2. Preparation and Characterization of Glycated Proteins

Hemoglobin prepared from fresh horse erythrocytes (TCS Biologicals, Wirral, Cheshire, UK) [93] was glycated in vitro by incubation with glucose using a slight modification of the methods previously reported [20,21]. Briefly, oxyhemoglobin (0.5 mM) containing 250 mM glucose plus 100 U/mL penicillin and 100 μg/mL streptomycin in Tris-HCl buffer, pH 7.5, containing 150 NaCl (TBS) was applied to Whatman grade 1 filter papers and incubated sterilely for 7 days at 37 °C. Un-glycated control hemoglobin was produced under the same conditions but without a glucose addition. Following incubation, the hemoglobin was desorbed from the filter paper in the above buffer and residual glucose and buffer constituents were removed by dialysis at 5 °C, firstly against deionized water, then against a 0.25 M ammonium acetate buffer, pH 8.5, and finally against the above buffer containing 20 mM magnesium chloride. The glycated fraction was enriched using phenylboronic acid (PBA) resin (Sigma-Aldrich) affinity chromatography according to Middle et al. [94]. PBA-bound glycated hemoglobin was desorbed from the resin by the addition of 1 mL of 0.25 M ammonium acetate, pH 8.5, containing 0.2 M sorbitol, and then dialyzed against deionized water at 5 °C to remove residual sorbitol.

The concentration of oxyhemoglobin, methemoglobin and hemichrome species in these preparations was calculated as described previously [74]. The degree of glycation of hemoglobin was determined using a thiobarbituric acid assay [63] and chemical modification was verified using LC/MS. For this purpose, 16 µM (with respect to tetramer) of the glycated and un-glycated hemoglobin samples, as prepared above, were dialyzed for 2 h at 5 °C against 0.1 M sodium phosphate buffer, pH 7.4. The hemoglobin samples were diluted 1:10 in the above buffer and incubated overnight at 37 °C with 0.5 µg of trypsin (Sigma-Aldrich). Samples were desalted by reversed-phase chromatography using ZipTips (Thermo Scientific, Whaltam, MA, USA). Aliquots of 0.2 µg were delivered into a Triple TOF 5600 mass spectrometer (Sciex, Framingham, MA, USA) by automated in-line reversed phase liquid chromatography (LC), using an Eksigent NanoUltra cHiPLC System (Sciex) mounted with microfluidic trap and analytical column (15 cm × 75 μm) packed with ChromXP C_18_-CL 3 μm (Sciex). Spectra were acquired automatically in positive ion mode using information-dependent acquisition powered by Analyst TF 1.5.1. software (Sciex). Protein sequence coverage was determined using ProteinPilot software v4.0 (Sciex). Glycated peptides identified by ProteinPilot were confirmed using PeakView software (Sciex) to extract parent ions of the appropriate m/z for peptide plus hexose (extracted ion count, XIC), and the area under the curve of each XIC peak was used to assess the presence or absence of the modified peptide in treated and untreated samples.

We chose type I collagen from bovine Achilles tendon as a representative substrate for glycation since type I collagen makes up approximately 70% of the total collagen in the dense connective tissue of the periodontal ligament. In vitro glycation of type I insoluble collagen with glucose was carried out as follows. In order to provide access of glucose to the otherwise insoluble tissue, we took advantage of the observation of Glimcher et al. [95] that insoluble collagen fibers can become hydrated by exposure to dilute organic acids such as acetic acid. To achieve this step, powdered bovine Achilles tendon (Sigma-Aldrich) was firstly suspended at a concentration of 1% (*w*/*v*) in 0.05 M acetic acid and allowed to swell at 4 °C for 48 h, with occasional stirring. Following this, the acetic acid in the suspension was neutralized by the addition of 1 tenth volume of 1 M sodium phosphate buffer, pH 6.8, and the collagen suspension was homogenized on ice using an Ultra Turrax (IKA Labortechnik, Staufen, Germany) tissue homogenizer for 5 min. To achieve glycation, we firstly added antimicrobial agents, sodium azide (final concentration 0.065% *w*/*v*) and penicillin and streptomycin to final concentrations of 100 IU/mL and 1 µg/mL, respectively, as described above. For glycation, powdered D-glucose was added slowly to give 1.25 M, whilst this was omitted from the control sample. Samples were then incubated at 37 °C for 35 days and then dialyzed against repeated changes of de-ionized water at 4 °C to remove buffer salts and glucose; the removal of the latter from the dialysis retentate was verified by assay using the Glucose Oxidase Activity Assay kit (Sigma-Aldrich). The preparations were stored frozen at −20 °C until required. To determine the degree of glycation, known volumes of the liquid samples were dried to constant weight at 40 °C, weighed and then assayed for 5-HMF as described above.

Un-glycated and glycated human serum albumin samples were purchased from Sigma-Aldrich (A3782 and A8301, respectively). The albumin-heme complex was prepared by incubating a 100 μM stock solution of human albumin in PBS at a 1:1.2 protein to heme molar ratio and subsequently passed through 7K MWCO Zeba™ Spin Desalting Columns (Thermo Scientific) [47,96] to ensure that no un-complexed heme remained. Albumin in complex with heme was analyzed using UV–visible spectroscopy.

Determination of secondary structure and heme-binding properties of un-glycated and glycated hemoglobin was carried out using near- and far-UV CD and UV–visible spectroscopies, respectively. CD analysis of hemoglobin was carried out on a Jasco 1100 circular dichroism spectrophotometer. Far- and near-UV data analysis was completed for the 180–260 nm and 260–320 nm wavelength regions, respectively. Experiments were conducted at 20 °C using hemoglobin concentrations of 0.5 and 5 µM (with respect to tetramer), for far- and near-UV analysis, respectively. Samples for analysis were first dialyzed against 0.1 M phosphate buffer, pH 7.4, to remove chloride ions and Tris, which otherwise would absorb at 180–200 nm, and were scanned 25 times and corrected against a control buffer. To analyze redox properties, 10 mM sodium dithionite (Sigma-Aldrich) prepared in PBS was used as the reductant [44,47,96].

Analysis of iron release from hemoglobin was carried out using the ferrozine assay method as described previously [97,98,99].

The efficiency of glycation of all proteins was also examined using sodium dodecyl sulfate-polyacrylamide gel electrophoresis (SDS-PAGE) and Coomassie Brilliant Blue G-250 (CBB) staining [47,96].

### 3.3. Analysis of Heme Release from Hemoglobin

To determine the ability of heme release from glycated and un-glycated hemoglobin by *P. gingivalis*, bacteria were grown in BM + Hm for 3 passages. Then, bacteria were centrifuged (4000× *g*, 20 min, 4 °C), washed once with PBS, and centrifuged. Analysis of heme release was carried out in the reaction buffer (50 mM Tris-HCl, pH 7.5, supplemented with 150 mM NaCl, 5 mM CaCl_2_ and 0.05% Tween 20) at room temperature, at a final bacterial culture OD_600_ equal to 0.15 and with hemoglobin added at a final concentration of 3 µM. Absorbance of the samples was measured at 406 nm (A_406nm_) and the heme release was shown as the decrease in A_406nm_ over time. Control samples omitted *P. gingivalis*. For each sample, a blank sample was prepared containing all components except hemoglobin.

### 3.4. Analysis of Proteins Susceptibility to Proteolysis

Total proteolytic activity of the wild-type *P. gingivalis* and gingipain mutant cells against glycated and un-glycated proteins was examined using SDS-PAGE. Bacteria were grown in BM + Hm for 3 passages. Proteins were treated with *P. gingivalis* cultures at an OD_600_ equal to 0.5. The reaction was carried out at 37 °C in 50 mM Tris-HCl, pH 7.5, supplemented with 150 mM NaCl, 5 mM CaCl_2_ and 0.05% Tween 20. At given time points, 30 µL of the samples was taken out and treated with a Protease Inhibitor Cocktail (Bimake, Houston, TX, USA), supplemented with 1 mM EDTA (Roth, Frederikssund, Denmark). The degradation of the proteins was analyzed by SDS-PAGE and proteins were stained with CBB as reported previously [47,96].

### 3.5. Protein Overexpression and Purification

HmuY protein was overexpressed in *E. coli* Rosetta (DE3)RIL cells (Agilent Technologies, Santa Clara, CA, USA) and purified from a soluble fraction obtained from *E. coli* cell lysate as described previously [47,96].

### 3.6. Heme-Protein Complex Formation and Analysis

Heme (hemin chloride; Fluka) solutions were prepared as reported previously [47,48,96]. Formation of heme-protein complexes was examined in PBS. UV–visible spectra were recorded in the range 250–700 nm with a double beam Jasco V-650 spectrophotometer using cuvettes with 10 mm path length. To analyze the redox properties of the heme iron, 10 mM sodium dithionite (Sigma-Aldrich) prepared in PBS was used as the reductant [44,47,96].

### 3.7. Heme Sequestration Experiments

Co-incubation of glycated and un-glycated hemoproteins with the purified HmuY was carried out in PBS at 37 °C and monitored using UV–visible spectroscopy under oxidizing conditions [47,95]. To demonstrate heme sequestration, hemoglobin (64 µM on a subunit basis) and albumin-heme complex (5 μM) were incubated at 20 °C with equimolar concentrations of HmuY. The UV–visible spectra were monitored at the indicated time points. The measurements were started immediately after mixing the HmuY protein with hemoglobin or albumin (time 0). The HmuY ability of heme sequestration from glycated and un-glycated hemoglobin was determined as previously described [98] by measuring the progressive increase in the area of the 525 and 560 nm Q bands of the HmuY-Fe(III)heme complex. The rate constant of HmuY-Fe(III)-heme association was calculated using a one-phase exponential association curve as previously described [98,100]. The HmuY ability of heme sequestration from glycated and un-glycated albumin was determined using difference absorbance spectra at 414 nm.

### 3.8. Biofilm Formation

The ability of *P. gingivalis* to form biofilms on the abiotic surfaces coated with bovine serum albumin (BSA; Sigma-Aldrich) or glycated and un-glycated collagen was examined as described previously with minor modifications [50]. Briefly, a 96-well plate was coated with collagen (200 µg/well) overnight at 4 °C. Then, wells were washed three times with PBS and unbound sites blocked with 1% BSA for 1 h at 37 °C, followed by washing three times with PBS. Wells coated with 200 µL of 1% BSA or un-coated wells were used as controls. Overnight cultures of *P. gingivalis* were centrifuged (4000× *g*, 20 min, 4 °C), washed once with PBS, and centrifuged. Bacteria were suspended in PBS to OD_600_ equal to 1.0 and 200 µL of bacterial suspension in PBS was added to the wells and incubated for 1 h at 37 °C. Then, wells were washed three times with PBS, cells stained with crystal violet solution (Roth) for 15 min, washed five times with PBS, and cell-bound crystal violet was solubilized by addition of 100 µL ethanol. Absorbance of all samples was measured at 570 nm.

### 3.9. Infection Assay

Immortalized human gingival keratinocytes (Gie-No3B11; ABM, Richmond, British Columbia, CA, USA) were grown in TM-040 medium (ABM), supplemented with 2% heat-inactivated fetal bovine serum (FBS; Cytogen, Zgierz, Poland), 2 mM L-glutamine (Cytogen), 100 U/mL penicillin and 100 μg/mL streptomycin (Cytogen) in a humidified atmosphere of 95% air and 5% CO_2_ at 37 °C in a CO_2_ incubator (Panasonic Healthcare Co. Ltd., Sakata, Oizumi-Machi, Ora-Gun Gunma, Japan). Gie-No3B11 cells were seeded at 1.0 × 10^4^/mL in 24-well plates for 24 h. Live *P. gingivalis* W83 cells were grown to the early stationary phase in BM + Hm, pelleted by centrifugation (4000× *g*, 20 min, 4 °C), and washed twice with PBS. Bacteria were added to keratinocytes cultured in DMEM medium (Sigma-Aldrich) supplemented with 2 µM glycated or 2 µM un-glycated hemoglobin without antibiotics in 24-well plates at a multiplicity of infection (MOI) of 100 and incubated for 4 h at 37 °C. Then, the medium was collected and spread on ABA plates in an appropriate dilution. To the remaining samples, fresh DMEM medium or medium supplemented with 300 μg/mL gentamicin and 200 μg/mL metronidazole was added. After 1 h, the cells were washed 3 times with PBS and lyzed to determine the number of live cells present inside the host cells. The experiment was carried out three times on two independent biological samples, each sample examined in three technical replications.

### 3.10. Bioinformatic and Statistical Analyses

A three-dimensional model of hemoglobin was constructed using protein structure deposited in RCSB PDB data base (PDB ID: 2MHB) [76] as a template and the Accelrys Discovery Studio 2016 Client (BIOVIA; San Diego, CA, USA) visualizer (now available as https://discover.3ds.com/discovery-studio-visualizer-download, accessed on 11 October 2021). A three-dimensional model of albumin was constructed using the protein structure deposited in the RCSB PDB data base (PDB ID: 1N5U) as a template and the Swiss PDB Viewer program [101].

The statistical analysis was performed using a one-way ANOVA test. In the infection assay, the two-way ANOVA test was employed. Data were expressed as mean values ± standard deviation (mean ± SD). For statistical analysis, the GraphPad software (GraphPad Prism 5.0 Inc., San Diego, CA, USA) was used.

## 4. Conclusions

In this study, we examined the properties of three commonly encountered proteins in the periodontal environment that are known to become glycated in vivo and that may serve as either protein substrates or easily accessible heme sources. We chose proteins differing in degrees of glycation (serum albumin being the faster), rates of protein turnover (collagen being the slower) and hemoglobin being the main heme source for *P. gingivalis*. Our results showed that the non-enzymatic glycation of two abundant proteins found in the periodontal environment, hemoglobin and collagen, might contribute to an increase in the pathogenic potential of *P. gingivalis* in vivo.

## Figures and Tables

**Figure 1 ijms-22-12084-f001:**
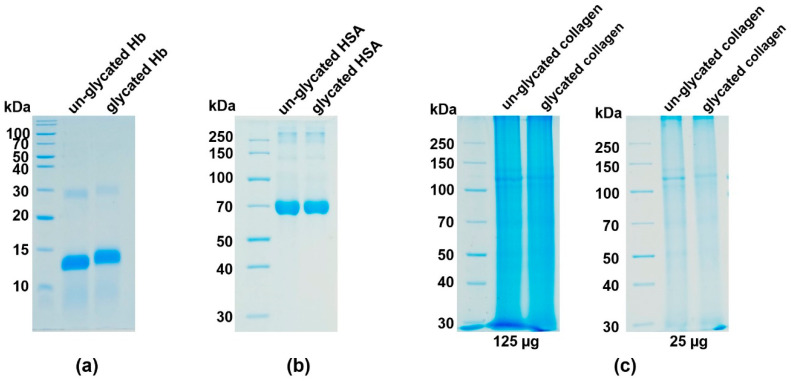
Analysis of glycation efficiency. Glycation of (**a**) hemoglobin, (**b**) albumin and (**c**) collagen was examined using SDS-PAGE and CBB G-250 staining. Verification of collagen glycation was carried out using samples containing 125 or 25 µg protein. Hb, hemoglobin; HSA, human serum albumin.

**Figure 2 ijms-22-12084-f002:**
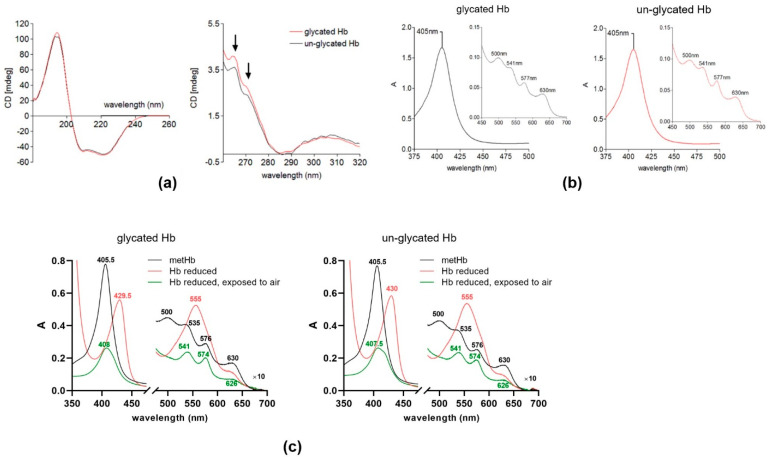
Characterization of glycated hemoglobin. (**a**) (Left) Far- and (right) near-UV CD analysis (averages of 25 repeat scans) of glycated and un-glycated hemoglobin. Both spectra were background-corrected by the subtraction of the buffer spectrum. Arrows in the near-UV region (260–280 nm) denote the wavelengths at which subtle changes were observed in the spectra of the glycated hemoglobin preparations. (**b**) The UV–visible spectra of the glycated (left panel) and un-glycated hemoglobin (right panel) after phenylboronic acid-affinity chromatography purification. (**c**) Comparison of redox properties of heme bound to glycated and un-glycated hemoglobin. Hb, hemoglobin; A, absorbance.

**Figure 3 ijms-22-12084-f003:**
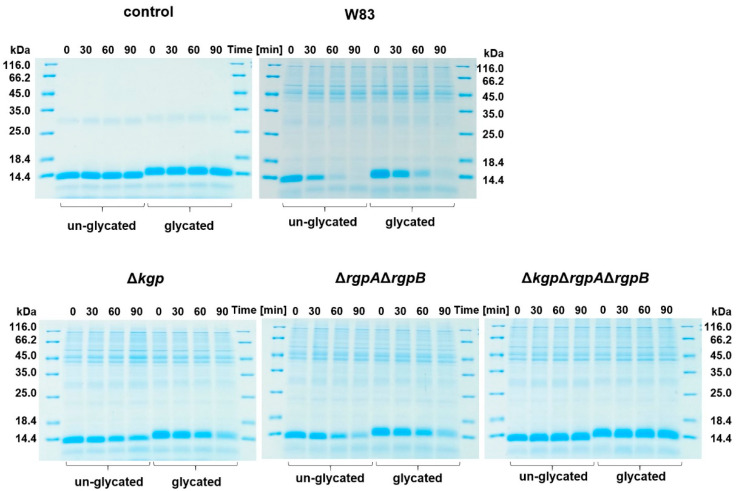
Susceptibility of hemoglobin to proteolytic activity of *P. gingivalis*. W83, wild-type *P. gingivalis* strain; Δ*kgp*, Δ*rgpA*Δ*rgpB*, Δ*kgp*Δ*rgpA*Δ*rgpB*, gingipain mutant strains lacking functional genes encoding lysine-specific Kgp and/or arginine-specific RgpA and RgpB, constructed in W83 strain.

**Figure 4 ijms-22-12084-f004:**
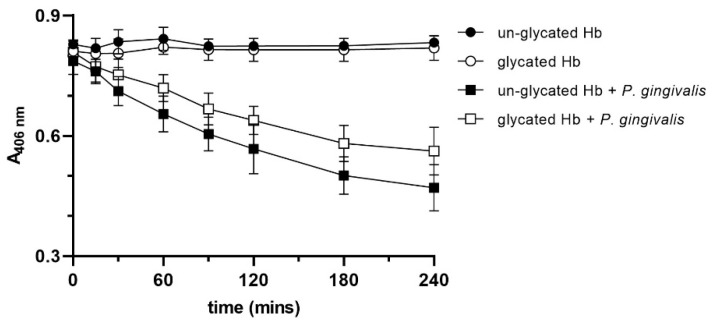
Heme release from hemoglobin after incubation with *P. gingivalis*. Absorbance of the samples was measured at 406 nm (A_406nm_), and the heme release is shown as the decrease in A_406nm_ with time. Control samples did not include *P. gingivalis*. The data are expressed as mean ± SD from three replicates.

**Figure 5 ijms-22-12084-f005:**
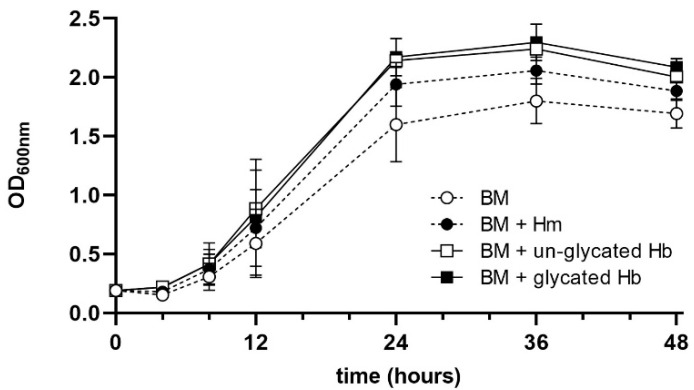
Analysis of *P. gingivalis* growth ability in liquid culture media. BM, basal medium; BM + Hm; BM supplemented with 7.7 μM hemin; BM + Hb, BM supplemented with 2 μM un-glycated or glycated hemoglobin; OD_600_, optical density of the culture measured at 600 nm. The data are expressed as mean ± SD values from two independent experiments, each performed in two replicates.

**Figure 6 ijms-22-12084-f006:**
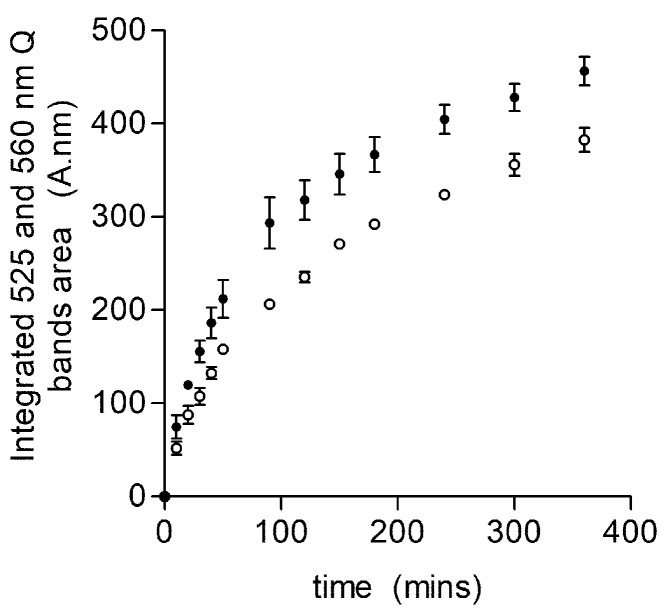
Comparison of HmuY-Fe(III)heme complex formation from hemoglobin. Glycated (close circles) and un-glycated (open circles) hemoglobin (64 μM with respect to monomer) samples were incubated with an equimolar concentration of HmuY. The combined integrated areas of 525 and 560 nm Q bands (from difference spectra) were used as a measure of HmuY-Fe(III)heme complex formation. The data points are an average from three separate experiments (mean ± SD).

**Figure 7 ijms-22-12084-f007:**
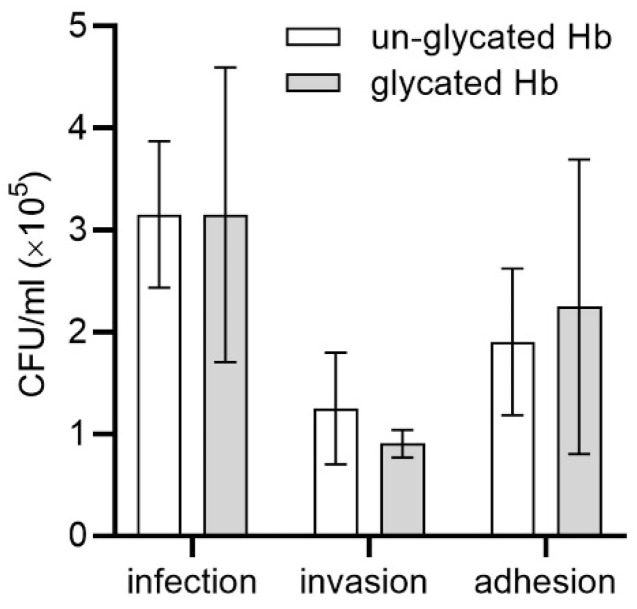
Infection assay. Immortalized human gingival keratinocytes were incubated with *P. ginvivalis* cells in the presence of glycated or un-glycated hemoglobin (Hb). The number of viable bacteria was determined as colony forming units (CFU) per mL. Adhesion, the number of live bacteria attached to the surface of keratinocytes; invasion, the number of live bacteria that invaded keratinocytes; infection, the total number of live bacteria in co-cultures of *P. gingivalis*-keratinocytes. The data are expressed as mean ± SD from one representative experiment showing a similar trend in three independent experiments.

**Figure 8 ijms-22-12084-f008:**
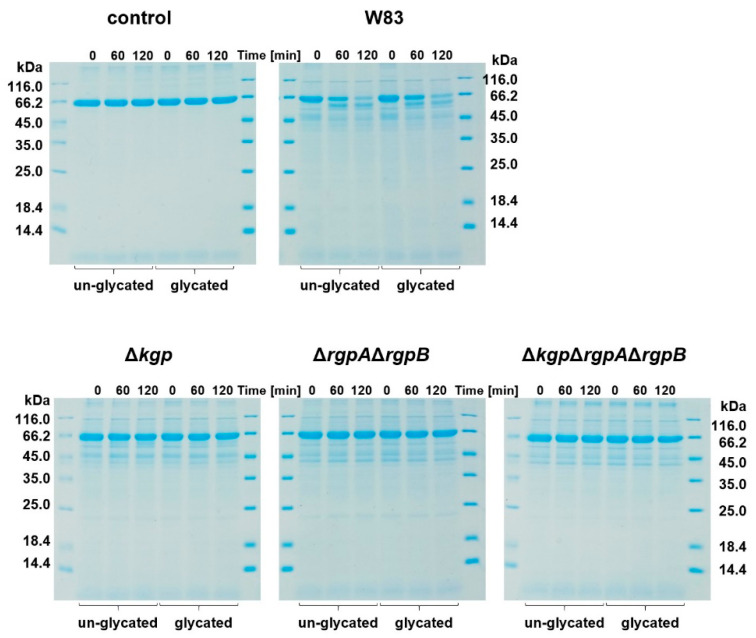
Susceptibility of albumin to proteolytic activity of *P. gingivalis*. W83, wild-type *P. gingivalis* strain; Δ*kgp*, Δ*rgpA*Δ*rgpB*, Δ*kgp*Δ*rgpA*Δ*rgpB*, gingipain mutant strains lacking functional genes encoding lysine-specific Kgp and/or arginine-specific RgpA and RgpB, constructed in W83 strain.

**Figure 9 ijms-22-12084-f009:**
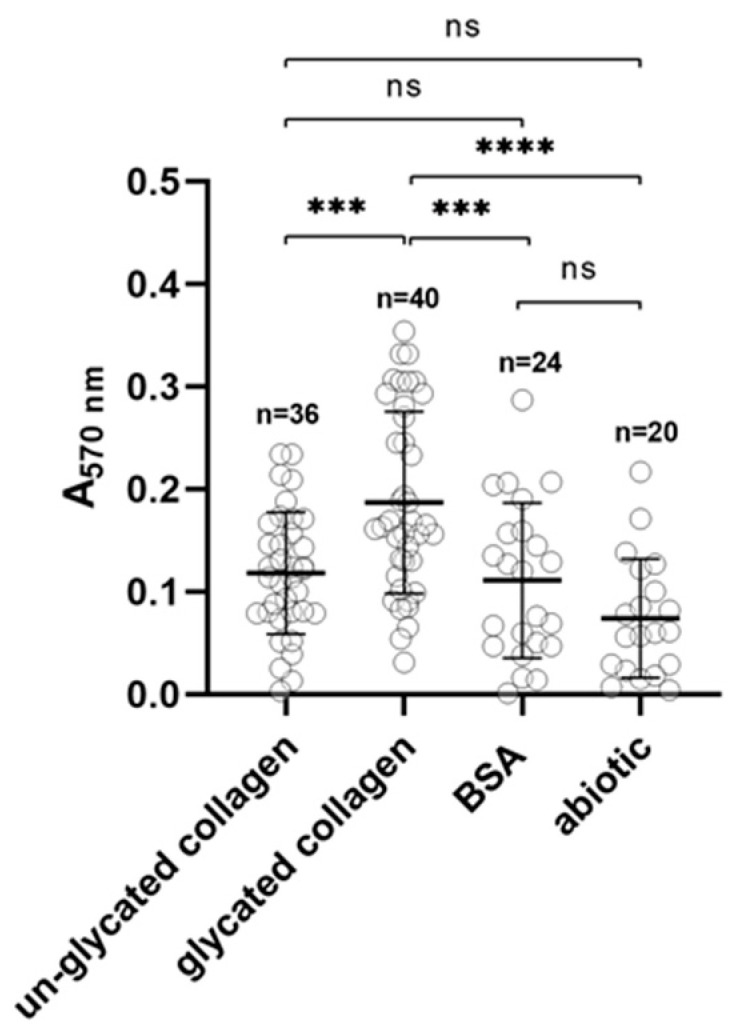
Biofilm formation. Ability of biofilm formation of *P. gingivalis* on glycated and un-glycated collagen-coated plates. Bovine serum albumin (BSA)-coated and un-coated (abiotic) plates were used as controls. *** *p* < 0.01; **** *p* < 0.001; ns, not significant; A, absorbance.

## Data Availability

Data are contained within the article, Appendix A and from the corresponding author upon request.

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
