# Peer review of "Glycation of Host Proteins Increases Pathogenic Potential of Porphyromonas gingivalis"

_ijms, 2021, doi:10.3390/ijms222112084_

Round 1

Reviewer 1 Report

Dear authors,

I find yor manuscript and design of the study very interesting.

Congratulations!

Please move the conclusion section at the end of the manuscript as presented in the IJMS template.

  1. Figures should be placed after they are first cited in the text, not before( figures 3,4,5)
  2. The conclusions section should present the main highlights of your study and normally does not include any references. Please rephrase this section
  3. It would be helpful to mention if there is any study in the literature on human subjects which detected the levels of glycated proteins mentioned here.

In conclusion, I consider it is a good article with interesting aspects.

Reviewer 2 Report

This manuscript by Śmiga et al. assessed the utilization of glycated host proteins by P. gingivalis in vitro. They generated glycated proteins, and test their function in growth, infection and biofilm formation of P.gingivalis. The results are interesting, but some parts need to be improved.

  1. Line 31-33, your current data did not support this conclusion. you should do statistical test to confirm the differences, and make the conclusion carefully.
  2. 1, this part should belong to introduction, not results part.
  3. Line 160-168, there should be a table or figure to show the data, not just describe your finding.
  4. To make the date easier to understand, I suggest you combine Figure 1 and Figure 2.
  5. Figure 4 and Figure 5, please show statistical method and whether they have significant differences. For Figure 5, did you do CFU to confirm this phenotype?
  6. Figure 7, how many samples you did? what is the statistical method? if there is no significant differences, your conclusion from line 320-324 should be removed. whether did you try other cell types?
  7. Figure 9, what is the statistical methods? please show up the repeat number.

Round 2

Reviewer 2 Report

Authors have addressed all of my questions.